# A Qualitative Study of the Spiritual Aspects of Parenting a Child with Down Syndrome

**DOI:** 10.3390/healthcare10030546

**Published:** 2022-03-16

**Authors:** Elysângela Dittz Duarte, Patrícia P. Braga, Bárbara R. Guimarães, Juliana B. da Silva, Sílvia Caldeira

**Affiliations:** 1Department of Maternal Child and Public Health, School of Nursing, Federal University of Minas Gerais, Belo Horizonte 30130-100, Brazil; b.radieddine@gmail.com (B.R.G.); juliana.barony@gmail.com (J.B.d.S.); 2Campus Centro Oeste–Divinópolis, Federal University of São João del Rei, Divinópolis 35501-296, Brazil; patricia_braga@ufsj.edu.br; 3Center for Interdisciplinary Research in Health, Institute of Health Sciences, Catholic University of Portugal, 1649-023 Lisbon, Portugal; scaldeira@ucp.pt

**Keywords:** beliefs, down syndrome, hope, family, parenting, spirituality

## Abstract

Parenting a child with Down syndrome can sometimes present certain difficulties and, thus, spirituality may function as a dimension related to finding meaning in life and as a coping resource. Spirituality is a critical dimension of nursing care, but scarce knowledge is available to specifically inform family nursing practice. The aim of this study was to explore the spiritual aspects of parenting a child with Down syndrome, as a qualitative secondary analysis. This is an observational qualitative study, based on in-depth interviews from 42 participants. Data analysis found seven categories that concern meaning and purpose in life: hope, family strength, spiritual practices, personal beliefs, and love, and trust in healthcare providers. Spirituality is a resource in parents’ lives who are living in this situation. Nurses should consider this dimension in supporting families and in improving management of this life and health condition.

## 1. Introduction

Spirituality has been studied as a resource to cope with potentially stressful situations [1,2]. Spirituality comprises the way in which each individual experiences moments of crisis, seeks meaning in life, experiences connections with self and others, and expresses beliefs, values, and traditions [3]. These are the main dimensions of spirituality as found in a concept analysis: meaning, connectedness and transcendence [4].

Spirituality can provide greater serenity to people experiencing any kind of need or disease. Spirituality influences coping with adverse situations and positively impacts quality of life [5]. In addition, spirituality is a supportive resource for patients and also for caregivers in overcoming stressful experiences.

Spirituality is broader than religion, which is related to a specific institutionalized belief and practices. Spirituality involves personal values, relates deeply to meaning in life and existence [3], promotes personal growth, facilitates reflecting on living experiences [6], and helps in adapting to new life conditions [7]. 

Spirituality is related to positive outcomes in health such as preventive factors of some diseases in the healthy population and shorter hospital stays in hospitalized patients [8]. The evidence on health outcomes is one major reason why spirituality is receiving more attention in healthcare in general, as well as in different healthcare contexts. For example, spirituality has been studied in cancer patients [9,10], in palliative care [9], with families of children with disabilities [11,12], with congenital heart disease [7], in gestational loss [13], in hospitalized children [14], bereaved parents [15], and infertile patients [16,17]. Each one of these situations has particularities as spirituality is a universal dimension but, at the same time, is lived individually and depends on specific situations that may challenge meaning in life and the individual’s belief system. Thus, families of children diagnosed with Down syndrome (DS) may have spiritual needs related to this challenging situation that involve family roles, parenting, and health. However, studies are still scarce on this topic, which may represent an opportunity to obtain more knowledge to inform nursing practice, based on a holistic paradigm. In this regard, families of children with heart disease indicated that spirituality deeply affected their lives and asked God for help to improve their children’s health, praying for the cure of their children [7]. In addition, some families understood their situation as a divine will, which facilitates accepting the disease, making faith a critical dimension to keep confident, calm and patient when facing distress. Promises, worship, prayer, and other religious practices were also found to be relaxing and able to give energy to deal with the child’s illness [7]. Cultural aspects were identified as important, such as religious ceremonies in China with the burning of incense and offering of gifts to ancestors [18], the belief in supernatural powers in Thailand [19], and the reading of the Koran [7].

In Brazil, a qualitative study conducted with families of children with chronic kidney disease identified aspects of faith and religiosity that helped families in overcoming anguish about the diagnosis, such as attending church, praying individually, and participating in a prayer group [20]. Brazilian mothers of infants with respiratory infection use folk medicine through the preparation of home remedies, medicinal plant teas, and seeking for *benzedura* (a cultural practice in Brazil aimed at healing the sick by applying ritualistic hands gestures, such as the sign of the cross, generally accompanied by a herb that is believed to have supernatural powers, at the same time with prayer) [21]. 

These are specific examples of living spirituality, particularly in a dimension related to beliefs that facilitate the search for meaning in life and promote comfort in situations of distress [1]. The birth of a child with DS may be an unexpected and a difficult event. The spirituality represents an opportunity to find more ability to accept and deal with this situation, which cannot be changed but requires lifelong commitment.

DS is among the health conditions related to disabilities and consequently demands healthcare across life. The estimated incidence of DS is approximately 1 in every 1000 live births worldwide. Every year, about 3000 to 5000 children are born with this chromosomal disorder [22]. In Brazil, the birth of 1 child with DS has been estimated for every 600 and 800 deliveries [23]. Both the specific therapeutic demands of these children and the global needs of the family make up a daily routine of demands and responsibilities [24]. In this process, the family can use different individual, collective and environmental resources to achieve good adaptation. In this regard, spirituality has been considered for a long time among the family and individual resources [25], even though the evidence remains scarce on this topic. Still, healthcare providers should consider spirituality as a resource in caring for these families [14,15,16,17,18,19,20,21,22,23,24,25,26]. NANDA International Incorporation, which is an international nursing diagnoses classification [27], includes nursing diagnoses related to spirituality, which reinforces the role of nurses in assessing and diagnosing spiritual answers of patients related to health or life processes or transitions. When nurses integrate spiritual assessment, they obtain deeper knowledge on patients’ wishes, and enhanced opportunity to plan effective interventions, according to patients’ beliefs and values, bringing a therapeutic and relational dimension into the process of caring and supporting, particularly for families of children with DS. It is assumed that spirituality is a resource for families who raise a child with DS and that it can contribute to the process of adaptation of the family to this situation. In this sense, this study seeks to answer the following question: How is spirituality, as a resource, perceived by the families of children with DS? 

If spirituality is not included for these families, then screening of distress and suffering may be compromised as well as the possibility of identifying strategies to facilitate the adaptation of the family to raising a child with DS. Thus, this study aimed to reveal how spirituality is described by the families of children with DS.

This study contributes to improving knowledge on how the spiritual dimension can influence fathers and mothers in caring for a child with DS.

## 2. Materials and Methods

The Consolidated Criteria for Reporting Qualitative Research guideline was used in reporting this study (COREQ) [28].

This is a qualitative secondary analysis (QSA) of the research project “Family and individual adjustment in Brazilian families living with children and adolescents with Down Syndrome” [29,30,31,32]. The QSA refers to using existing data collected for a different research work [33], allowing a new perspective or look at the original data. An analytic expansion of QSA was used in this study [34]. In the primary study, the spirituality of families of children with DS was not the focus of the interviews. However, participants presented consistent information about this theme, especially when talking about social support, problem solving, and coping strategies. From a qualitative perspective, this information was considered important in the life context of families living with a child with DS and a secondary research question was defined aiming at this specific theme. The primary data were useful in asking new or emerging questions derived from the initial research question. This QSA was subsequent and the time from the primary study to this end has brought opportunities for new insights to researchers, which is described as an advantage of this procedure [35]. 

The primary study was an observational, qualitative study conducted between February and November 2017, with 39 mothers and three fathers of children aged between 1 and 7 years, diagnosed with DS, who were residents in two cities in the state of Minas Gerais, Brazil. The exclusion criteria were caregivers’ impaired communication or psychological status that compromised the ability to participate in the study. 

Participants were first identified and then recruited by healthcare services, non-governmental organizations, and healthcare providers who usually care for children with DS and their families. No database related to individuals with DS is officially available in Brazil and this was the strategy to find participants. Then, a snowball technique was used to find more families that fitted the inclusion criteria. Data collection was performed using semi-structured, face-to-face interviews with questions based on the theoretical framework “Model of Resilience, Stress, Adjustment and Family Adaptation” [36]. Questions addressed aspects related to the family description of having a child with DS, family resources, problem solving, access to information about the condition, and previous knowledge of DS. The following questions allowed information to be obtained related to spirituality and were asked to families to explore family resources in the primary study: “What were the strategies you used to face the moment of diagnosis?” “What or who helped you at that time?” “I would appreciate if you would tell me what it was like for you and your family to take care of your child in the early years of their life?”, “In the future, what do you think the impact of having a family member with Down syndrome will be like? All interviews were conducted by one researcher with qualitative research training. The interviews lasted an average of 74 min and 55 s. Two researchers transcribed the interviews that were audio recorded, and compared the audio with the transcripts for accuracy. Each participant was identified by a code composed of participants’ condition (M if mother, F if father, C is child) followed by a number according to the sequence of interviews (e.g., M2, F8…). MAXQDA 2018 software was used to treat the data. For the primary study, the researchers realized direct content analysis. Four main codes were identified: (i) The arrival of the child with DS; (ii) Family demands and challenges in the ongoing care of the child with DS; (iii) Family assessment of the situation and the stressors; and (iv) Family resources, problem solving and coping strategies. Concerning this last code, when talking about family resources, thirty participants of the original study referred to aspects related to spirituality. These findings have disclosed the importance of this resource for the families of children with DS and provide the rationale for conducting further specific analysis.

The research was approved by the Research Ethics Committee according to opinion CAAE: 1039746614.9.0000.5132. It is worth emphasizing that the current study was conducted in compliance with National Health Council Resolutions 466/12 and 510/2016 [37,38], which concern the guidelines and standards recommended for human research. All participants freely provided informed consent. 

For this QSA, there were 30 interviews from the original database that had information on spirituality. All these interviews were subjected to thematic analysis in a deductive approach [39]. The researchers returned to the transcripts of the primary study and selected the quotations guided by the concept of spirituality based on Narayanasamy (2010) [3]. Initial codes were created based on the theoretical framework; a final code system with seven codes was reached after review and discussion. A codebook with a detailed description of each code was created to support the coding process. Two authors (BRG and EDD) independently coded the first five interviews and discussed disagreements for training purposes to ensure appropriate coding and definition in situations of divergence [40]. When an agreement between the coders was verified, the interviews were coded by only one of the researchers. The agreement was verified through interrater reliability using the MaxQDA, achieving k scores > 0.75. The similar codes were grouped into themes and organized to answer the research question. The themes created were: (1) Meaning and purpose in life of children with DS; (2) Hope and family strength; (3) Trust and connection; (4) Spiritual practices, the concept of God or other higher Being (Divinity); (5) Personal beliefs and values; (6) Meeting the needs of love; (7) and Contribution to the transformation in life.

The processes of the primary study and the QSA are demonstrated in Figure 1. The final themes, their definition, and sample quotation are presented in Table 1.

## 3. Results

The age average of participants was 40.17 years old and, for children, 3 years old. Parents’ ages ranged from 19 to 49 years old, and children’s ages ranged from 1 to 7 years old. The respondents were mostly women (92.8%), living with a partner (88.2%), graduated (52.5%), and had a salary higher than BRL 5000 (Table 2).

### 3.1. The Meaning in Life of Parents of Children with DS 

This category was built from the meaning and purpose of life that the participants recognized as the reason for the birth of a child with DS in their family. During adverse situations, the search for meaning and purpose in life is the primary force in discovering a meaning for suffering. This is a critical aspect of spirituality and suggests that people with a sense of meaning and purpose in life cope better with difficult circumstances [3].

The results revealed that childcare was expressed as a “mission” to be fulfilled. Those who are close to these families recognize the care of children with DS by their parents (M13, M15 and M7) as a life purpose and define them as “very special people” and “chosen parents”. This purpose is taken by the participants with intensity and can be seen in the speech of F7 who declares feeling “victorious for having him (her child) [C7]” and that “if he has to have C7 30 million times in another life, I will”.

“[…] one thing that people passed on to us who said that ‘if you received a special child, it’s because you are special. God wouldn’t give a child to someone who isn’t special’” [M13]

“The reaction of people was very interesting, you know, “wow, you are very special” that irritated me a little at first too… “because you are very special, that’s why God gave you a special son… now I I understand that… you really have to be very special because it requires immense dedication.” [M15]

### 3.2. Hope and Family Strength 

This theme refers to situations experienced in the care of children with DS that contribute to the strengthening of the family as a whole and promote hope regarding the future of their children with DS. Hope enables participants to recognize the possibilities in their life and in the life of the child with DS, strengthening to create different methods of coping. Therefore, participants described strength and hope as being related to concrete actions. Hope and family strength was understood as the resources sought by the family for its strengthening, with emphasis on the participants’ development to understand the particularities of their children and the search for specialized assistance.

Projecting a future for their child from the moment they learned of the diagnosis was one of the factors identified in the discourse of F7 that allows recognition of the existence of hope in the face of the situation experienced. Even with the geneticist information that “they could not create expectations” regarding the child’s future, F7 states that he did not stop thinking about the child’s future. He also states that he recognizes “his limitations, that his time is not the same as the time I want it to be” and that he believes that his expectations for the child’s future will be fulfilled according to “God’s time”.

“I’m going to create expectations in my son, yes” “ah but, no father, no, I didn’t mean that” I said like “no, but wait, yes, I’m going to create, but I know his limitations, I know that his time is not the same as the time I want it to be” did you understand? And nobody’s time is the same, isn’t it? Is God’s time tended? God’s clock is different from ours. We want ours for today and it’s not, it’s for tomorrow, it’s for two months, three…” [F7]

The hope of the child’s development and the achievement of skills, such as literacy, can be identified in the speech of M14 when she says that “if everyone can [be literate], she will also succeed. She’s going to talk, she’s going to walk, she’s going to read, at the right time she’s going to do it”. It is noteworthy that, similarly to what was previously presented in a report by F7, maternal expectations take into account the uniqueness of her child, who may have a different time to achieve what is expected.

Although her child is still a child, M15 speaks of the hope that her child will have autonomy in the future, with the ability “to work… to support himself, to take care of his own personal things… even arrived… I don’t know if you make a meal, something like that…but why not? Right?” (M15). In order to achieve this goal, she has sought care in institutions that can contribute to the child’s development, and M15 has also sought information to help her improve the care she offers her child.

### 3.3. Trust and Connection 

Trust refers to the sense of security that can be established in relationships with others and is essential for spiritual health and a sense of wellbeing [3]. In this study, these were identified by participants when referring to their feeling towards other people in the family, community and healthcare providers. It is evident that the recognition of trust in these people stems from their contribution to the care of children with DS or from the support offered to the study participants so that they could continue with this care.

The sources of trust mentioned by the participants were the parent associations (M14, M16, M31, M37, M32, M23), health professionals (M15, M17, M37, M32, M22, M23), the partner (M7), close relatives (M7, M16, M22, M23, M32) and friends (M23).

The associations of parents and professionals were especially important in providing information for the care of children with DS and to enable assistance in healthcare services. It is evident that, in addition to the professionals’ ability to contribute to care, the personal characteristics of these professionals, such as kindness, attention, and communication skills, were deeply appreciated. Trust is expressed when referring to the support received for care, the affection offered, the exchange of information and experiences.

“The life of C31 changed a lot after we got in contact with the Parents’ Association 1.” [M31]

“Doctors guide me a lot, like, go to the pediatrician, she guides me, the physiotherapist, oh God! […] she was my safe haven.” [M17]

“When C7 was born, we were very united. […] sometimes F7 [partner] caught me crying, and hugged me a lot, you know?” [M7]

“Oh, it was… I, in my case, it was the support that my children have given me, right…” [M17]

“What helped the most was the support of the family, the support of God especially and the support of my husband.” [M32]

### 3.4. Personal Beliefs and Values 

Personal beliefs and values are what people place the greatest value on in their lives and drive their faith. Personal values can be beliefs of a religious affiliation, a set of philosophical statements or a physical activity or new leisure that they believe contributes to their lives [3].

The participants who believe in God recognize him as a source of courage to face the challenges of having a child with DS and the one who, as reported by M18, “provides everything”. Even wondering “why did I have a special child? Why did God do this to me?”, M19 recognizes God as a source of courage when he says that “if it wasn’t for Him I wouldn’t have, you know? I wouldn’t face it.”

“…And thank God, I have to thank…I’ll tell you this, when we have faith in God, we don’t need to fear, because God provides everything…” [M18]

“Jesus, Jesus, I have great faith, that’s Him, I think if it wasn’t for Him I wouldn’t have that/you know? I wouldn’t face it, because there’s a lot of questions that come to our minds “why did I have a special child? Why did God do this to me?” Honestly, mothers have these questions a lot. I thank God, although it is logical that I also have these questions, but you know, I believe that the Lord has a transformation to make in her life, in my life, you know? That’s why I feel calmer”. [M19]

Belief in God and in his ability to determine the course of their lives was expressed by claiming that “the Lord has a transformation to make in her life” (M19), that the child was “given to God” and that “lives by God” (M18). It appears that the belief they have is capable of providing tranquility and withstanding the adversities resulting from care.

“Honestly, mothers have a lot of these questions, I thank God, although it is logical that I also have these questions, but you know, I believe that the Lord has a transformation to make in her life, in my life, you know? That’s why I feel calmer”. [M19]

“Ah! Faith, here at home, there is a lot of this side, I think that faith comes first” [M24]

“Only God! The faith!” [M37]

### 3.5. Spiritual Practices, the Concept of God or Other Higher Being (Divinity) 

This category was defined from the participants’ statements expressing spirituality through rites and religious practices that provided them with a connection with the Divine or Sacred. Therefore, it concerns their experiences of communion with the transcendent, prayer practices and rituals.

Participants reported spiritual practices such as religious rituals, prayer, meditation, going to church and reading the bible. These practices were recognized as dealing with the situation of having a child with DS.

Participants’ belief in God and in his ability to determine the course of their lives is evident when they affirm “the Lord has a transformation to make in her life” (M19), that the child was “given to God” and “lives by God” (M18). It appears that the belief they have is capable of providing serenity, helping to withstand the adversities resulting from care.

“Jesus, Jesus, I have great faith, that’s Him, I think if it wasn’t for Him I wouldn’t have that/you know? I wouldn’t face it” [M15]

“[…] It was God who supported me, all the time. And when you have God at the helm, he uses good people to guide you…” [M18]

“He [God] gives me the condition to create” [M29]

“[…] God enabled me… God gave me strength” [M18]

“[…] My faith, I prayed a lot in his little legs, in his little body, one year and seven months, it’s a testimony of life, you know?” [M24]

### 3.6. Meeting the Needs of Love

The data allowed us to identify people who are a source of unconditional love and opportunities to experience love as a harmonious and lasting relationship, regardless of failures, gains or actions [3]. For this theme, the segments referring to people who contributed to meeting the needs of love of the participants and/or the child with DS were considered.

Family members are identified as people who meet the child’s needs for love, as is the case with M14 when stating that “I love her regardless of her having syndrome”; the other children of M15 who, upon learning of the possibility of her brother having DS, said “it’s my brother anyway, we’re going to love anyway”. The existence of love to meet this need was also identified by the participants in other close relatives (M17, M22) and in healthcare providers (M15). Most of the situations of unconditional love reported by the participants have already been identified by them even before the child’s birth.

“You idealize, right, one thing and you receive another, right. […] You love the same way. […] Whether she has syndrome or not is a detail. I won’t forget about the syndrome, no, I won’t talk like these mothers say no, but I love her regardless of whether she has syndrome or not. She is my daughter, right, I was the one who generated her and she is mine…” [M14]

“My mother, she drowned at 49 years old, 22 years ago she became quadriplegic…and so, she is hallucinated with the C15 and he with her…she asks to go to the bedroom to see…they interact, I think it’s really cute… he blows a kiss, when she’s sleeping he’s like “hey, hey, hey” wanting to wake up, you know?” [M15]

“When I broke the news to my children, ‘o mother, it’s my brother in the same way, we’re going to love in the same way…’, so that, I think it gave me more strength too” [M17]

“He loves being a father so much, you know?! So he is very happy to be a father! The same, from time to time he comments to me like this: “Ah! I don’t even remember that C22 has Down Syndrome”, you know?!” [M22]

“(When M7 received the diagnosis) My husband even thought I was going to leave my son, I said “I would never do that in my life, would you?”. Regardless of what he had, if it was cerebral palsy, if it was down syndrome, autism, whatever, he’s my son, and I, whatever I can within my conditions, I’ll give everything I can to stimulate and support it tended?” [M7] 

### 3.7. Contribution to the Transformation in Life

The experience of caring for a child with DS allowed participants to recognize the contribution of these children to transform the lives of other people who live with them. This theme addresses the quotations that present the benefits perceived by participants resulting from living with the child, the new meanings found in their lives and the family group, and the learning obtained from the experience of participating in care.

Positive contributions are identified as “maturity” (M18, M40), the ability to “see life lighter, you know, calmer, more connected to God” (M40), greater family union (M22, M33 and M25), feeling “very loved by my friends, by my family members” (M22) and the elimination of prejudice towards people (M10) “if I had few prejudices before, now I have almost none. And I really don’t.”

“No…no…I think if it had been different, maybe I wouldn’t be that person, and I wouldn’t have this maturity with everything the way I have today” [M18]

“Ah more maturity, you know, to face life in all the difficulties, in all of them, regardless of whether we’re talking about the C40 […], everyone is more mature, everyone is seeing life lighter, you know, more calm, everyone more connected to God” [M40]

“I felt very loved by my friends, by my family, everyone visited me a lot, you know? They were very affectionate with me, everyone was very affectionate with the C22, so what I can say in terms of the family is that everyone came together more, you know?!” [M22].

“What I think happened is that we got even more united, you know, like all for one” [M33]

“I’m a mother now, I’m no longer the M10 I used to be…now I’m a M10 mother, right, it’s different… and the issue of values changes a lot, especially in the case of having a child with a disability […], if before I had few prejudices, now I have almost none. And I really don’t” [M10].

“(When C14 was born) I didn’t want a visit, I didn’t tell anyone I won, I had… you know, that. […] it’s a mourning that… I don’t even know how to explain it to you. Today, after it passes, we can talk, but it’s a mourning that you have no idea about… it was very difficult, very difficult, ok? And still. It doesn’t go away like that, and I don’t think it ever goes away. Don’t pass it, you carry it for the rest of your life. This… this pain, which is really a pain, you know?” [M14].

## 4. Discussion

The analysis shows that the experience of having a child with DS is described by parents and families as if they were chosen and as a mission of caring for these children, perceived as a life purpose. The search for meaning in this life condition is a primary force in life that was intensified. People who seek meaning and purpose for their lived experiences can adapt more easily to difficult circumstances [3]. For this, some families strengthen their support network and intrafamilial bonding, recognizing the singularities of a child with DS [41]. 

Upon being informed, the diagnosis of DS can cause shock, denial, guilt, and anger and it is common for these parents to go through a process of adjustment to the new reality. Research conducted with families of children with congenital malformations [42] and microcephaly [43] also showed feelings of shock, crying, and grief when the expected child was not born, as well as a certain degree of rejection of the newborn child. Even so, as time went by, they saw themselves as the only people responsible for the care and the political role of fighting for their children’s rights [43]. Thus, they saw their life mission as “being a mother”, which became a gift, a new political occupation, and a kind of maternal “profession” [43]. 

Therefore, the preparation of the family for childcare implied reorganization and acquisition of new knowledge and practices. These aspects were recognized by the participants as moments of becoming more mature and developing problem solving skills. The set of skills needed to care for the demands of the child was also reported as transformative in the lives of these families. 

However, by unconditionally dedicating herself to the care of her child, the mother, often as the primary caregiver, becomes vulnerable to physical and emotional overload [44]. In this sense, basic daily activities, such as feeding, sleeping, and resting, are impaired, interfering with their quality of life. The search for power or a superior being may support them to find satisfaction in the search for meeting other life needs. 

Faith in God is revealed as an expression of personal belief and a resource that strengthens hope and helps to face daily challenges. This connection with the transcendent is also expressed in other studies that relate it to increased hope, feelings of gratitude, positivity, and a source of strength [45]. These aspects help families deal with the difficulties arising from the chronic condition and find meaning in their lives. 

A Brazilian study conducted with parents of children with DS, between one to seven, years old, showed that belief in God was referred to by the participants as a resource for them to adapt to the fact of caring for a child with DS [29]. Other studies conducted with children with chronic conditions also showed faith in God as a support in times of difficulty and support to cope with the demands arising from health problems [11,46]. 

Data analysis shows that spiritual practices such as religious rituals, prayer, meditation, and Bible reading were reported by participants. In what concerns religiosity, individuals may experience their existence not only as a task but as a mission determined by a higher being [3]. In Brazil, the main religions are Roman Catholic, Evangelical, Spiritualist, and Jehovah’s Witness, respectively [47]. Religion has helped people cope with crises, offering them understandings about suffering, ways to re-signify their distress and improve the quality of life of patients and their families [45]. Thus, religious practices can help people deal with their physical, psychosocial, or spiritual struggles. 

Participants referred to God as support and transformation, always providing improvements in the lives of families, and providing the necessary resources to face their difficulties (M15, M18, M24, and M28). They also consider that there is a purpose of God for them to be exposed to this situation. Among these resources is the strength and capacity they acquire to take care of the children and the meeting with people willing to help in the care needs. 

Thus, meeting the spiritual needs of an individual is related to the quality of life throughout the trajectory of the experience of caring for the child with DS. In this sense, cure-oriented care should be directed to care that promotes the family’s well-being, also contemplating actions focused on spirituality [45]. Studies indicate that nursing interventions can significantly improve the level of hope among patients with different chronic conditions. Care provided by nurses has been suggested to keep and encourage hope in these families [48]. A study with nurses working with children in palliative care showed that these professionals are in a better position to recognize the beliefs of the child and the family, their sense of spirituality and plan strategies to help in accepting and adapting to the child’s new health condition [49]. 

A Brazilian survey conducted with 27 healthcare professionals aimed to investigate how the multi-professional health team deals with the spiritual dimension of hospitalized patients and found that the smallest gesture provided with love allows for dialogue, improves thinking about faith and positivity to treatment, according to patients’ perspectives [50]. Spirituality has been highlighted in the health field because it represents a source of strength and comfort at difficult times in people’s lives. Therefore, healthcare professionals cannot neglect spiritual care and this should be offered without judgment or imposition of religion, since spirituality may or may not be related to formal religious practice.

Participants’ reports show the importance of bonding and support from people around, whether family, friends, neighbors, or health professionals. The relationship with healthcare professionals is useful to outline strategies that can solve or minimally allow them to actively deal with this family’s problem. In the case of caring for a child with DS, having informational support from the health professional increases the sense of competence, future planning and indicates an active coping with the stressful situation [51]. 

In this sense, in addition to the guidance provided on the child’s clinical condition, the healthcare professional must use strategies to investigate and promote the family’s feeling of hope, and also be aware that it is better to be prepared. By identifying and encouraging the family to build feelings of hope and future plans for the children, the nurse may contribute toward nurturing spirituality. In addition, it is appropriate to explore and listen carefully to the questioning, meanings, and purposes assigned by families regarding DS in their lives [49]. Valuing and recognizing this dimension of spirituality will make sense to them and may contribute to them feeling cared for [3]. 

In this research, healthcare professionals were important, especially in offering information for the care of the child with DS and to enable care in healthcare services. It is evident that the personal characteristics of these professionals were emphasized, such as kindness, attention, communication skills, commitment to ensure the necessary care for the child and the ability to recognize the mothers’ care needs as well. 

In addition to healthcare professionals, families also mentioned the support offered by parents’ associations that share similar experiences. It was observed that this contact allows for a feeling of safety and a more remarkable ability to care for their children.

Family members are seen as being able to establish a relationship of trust with each other [3], being able to support care, affection, and the exchange of information and experiences. In situations of adversity such as having a child with DS, the establishment of relationships of trust is fundamental for coping and sharing demands, insecurities, and daily doubts experienced by families. This support network contributes to the family’s adaptation to the condition of having a child with DS [52]. A study that aimed to find out how fathers and mothers feel about having a child with SD showed that over time, these families report benefits, such as family unity, experiences of love, affection, and pride with the child, in addition to expressing greater joy, patience, and tolerance in their lives [51]. 

The need to love and be loved are fundamental and spiritual needs [3]. Participants stated that the syndrome did not modify the families’ love for their children. A study with 58 parents of children with a chronic condition relates the feeling of love for their children to the aspects: affection, acceptance, sensitivity, care, and support [53]. According to Narayanasamy (2010) [3], people manifest love needs when they show feelings such as self-pity, depression, insecurity, isolation, or fear. Thus, these are situations representing lack or need for love that should be met. 

Love was expressed as an unconditional feeling, even before living with the child with DS. Few studies are available that investigated the dimension of love in relation to caring for a child with a disability. However, from the findings up to this point, it is possible to infer that the feeling of love brings fathers, mothers, and other family members together for the care focused entirely on the child’s unique health needs. In addition, love can turn a situation of high demand for care and overload of tasks into a rewarding “life mission” situation. 

It must be considered that the period of development from the child with DS into adulthood is permeated with uncertainty [54]. Although there are advances to help improve cognitive function, behavioral function, and quality of life in adults with DS as they age, there is still a great deal of uncertainty regarding the future of the child with DS [55]. This reinforces the need for families to build hope, that is, expectations and feasible future plans for the child’s condition. This future-oriented hope allows for feelings of peace and contentment and a positive perspective of what is experienced [3]. It is thus considered that hope contributes to families being able to envision the development of their children and a future with possibilities so that children can acquire autonomy in their lives. 

A study that sought to analyze hope in mothers of children with DS and its association with coping behaviors adopted by the family indicated that belief in God or religiosity were significantly associated with greater hope and relationship quality among parents of children with DS [56]. The data from this research also present the belief in God and the hope that their child will have a satisfactory development in the future. 

In this sense, spirituality is related to the capacity of this family to feel hope for a future of possibilities for the child. In addition this belief, focused on the future and grounded in faith, family and healthcare providers help these families to define actions and strategies of encouragement and persistent stimulation for the child, in the search to reach milestones in global development. 

## 5. Conclusions

This study concerns a secondary analysis of the adaptation of Brazilian families living with children and teens with DS. Spirituality has been described as a critical coping resource in times of crisis and several spiritual aspects have been found in this study that were disclosed in parents’ segments from interviews. Spirituality is a deep dimension of life, related to values and beliefs, essential in providing meaning, a sense of hope and love, and nurturing personal growth. The study confirms the findings of previous research on the importance of spirituality in healthcare, and brings evidence on this specific population. Using a theoretical framework to analyze the data was critical in studying this topic. Spirituality emerged in many different components, which included religious and non-religious aspects and were concrete and contextualized in the daily life. Family nurses and all those providing care for families with DS should be prepared to assess spiritual needs and implement spiritual interventions. This specific context offers a quite innovative opportunity to provide spiritual care and effectively implement a holistic and family-centered approach. 

## Figures and Tables

**Figure 1 healthcare-10-00546-f001:**
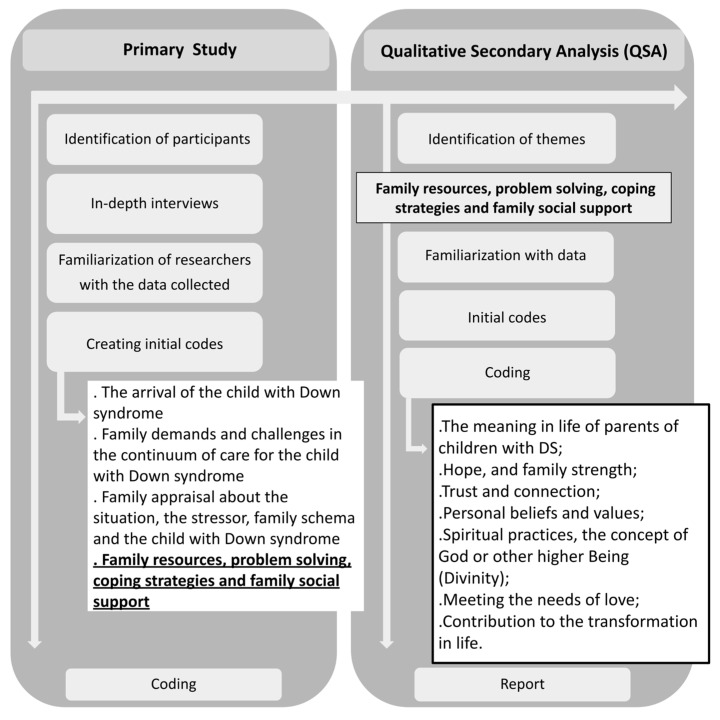
Flowchart of the work process of the original study and the QSA.

**Table 1 healthcare-10-00546-t001:** Themes, definition, and illustrative quotes.

Themes and Definition	Illustrative Quotes
The meaning in life of caregivers of children with DS—the meaning and purpose in life that the participants recognize and that, from their perspective, explains the reason for the birth of a child with DS in their family.	Caring for children with DS is a life mission
	“[…] I have always wondered what is the purpose of my life? What is my life goal and mission? As so, this is one of my answers (to care), this is my son” [M7]
Hope and family strength—every kind of situation lived by the family that globally strengthen the family and nurture hope towards the future of yourchild with DS.	Thinking of the future with optimist but keeping the awareness of limitations “I want to raise C15 as a child…provisioning conditions, an adult that has a job, that is able to keep the ability to take care and self care, such as cooking…I don’t know…Kind of…why not?’’ [M15]
Trust and connection—feeling of trust reported by the participants towards other people in the family, the community and health professionals and who somehow contributed to the care of their child with DS.	Parents associations
	“Life of C31 has changed a lot after we’ve been in touch with Parents Association 1” [M31]
Personal beliefs and values of caregivers—aspects identified by participants that is valuable in life and faith, whether or not related to any religion.	Belief in God
	‘’I believe in God, my faith is big, and God is like that, I guess that if not God I would not have this…you know? I would not be able to deal with all this situation, because there are so many questions within our mind: why me, God? Why I have delivered a special child” [M19]
Expression of spirituality—situations described by the participants who expressed their spirituality through rites and traditional religious practices or rituals performed to facilitate a connection with the Divine or Sacred. These expressions include religious or non-religious topics.	Believe in God and in God’s will
	“God provides the strength and removes obstacles, without God I would not be able to live this, my strength comes from God, you know? As so, my faith…I usually pray in his legs, his body. And this has been going on for 1 year and 7 months! This is a testimony of life, right? ” [M29]
Meeting the needs of love—considered loving relationships, harmonious and lasting relationships that the participants reported in relation to other people and that met the love needs of the participant and/or their child with DS.	Parents and love for the child with DS
	“You imagine something, right, and you get another different one […] You love at the same way […] Having a syndrome comes to a detail I won’t forget the syndrome just like some mothers do, but I love her having or not the syndrome.” [M14]
Contribution to transformation in life—situations described by the participants in which they recognize their contribution to transforming the lives of others who live with them, since the child was born, but also the transformation in their own lives.	Benefits perceived by the family as whole
	“I guess we all became closer and it is all in one” [M33]

**Table 2 healthcare-10-00546-t002:** Characterization of parents and children with DS, in Belo Horizonte City, Minas Gerais State, Brazil, 2021.

Variable	Mean	Standard Deviation
Age (average) of child	3.02	1.88
Age (average) of the person in charge	40.17	7.56
**Variable**	**n**	**%**
**Gender (*n* = 42)**		
Father	3	7.2
Mother	39	92.8
**Marital status (*n* = 42)**		
Married	36	85.72
Divorced	2	4.76
Separated	2	4.76
Widowed	1	2.38
Stable union	1	2.38
**Schooling (*n* = 40)**		
Elementary school incomplete	2	5
Elementary school complete	2	5
High school complete	13	32.5
Higher education incomplete	2	5
Higher education	21	52.5
**Monthly family income in BRL (*n* = 40)**		
<1000	5	12.5
1001 to 2000	8	20
2001 to 3000	5	12.5
3001 to 4000	4	10
4001 to 5000	6	15
>5000	12	30

## Data Availability

No publicly archived data were used.

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
