# Peer review of "A Qualitative Study of the Spiritual Aspects of Parenting a Child with Down Syndrome"

_healthcare, 2022, doi:10.3390/healthcare10030546_

Round 1

Reviewer 1 Report

The present manuscript reports results from interviews with families who have a child with down syndrome. The focus of the interviews (or the results, which I cannot discern based on the text) was on spiritual aspects of coping with this life situation.

Before I go into detail, I want to disclose that I am not an expert in qualitative research. However, as some close colleagues of mine are doing qualitative research, I know something about qualitative methodology.

In the present paper, the research objectives are too little elaborated. It is getting clear from the first few paragraphs that the authors attribute spirituality a major role in coping with wearing life situations. They report studies that have shown that before. The results are presented in a manner that leave no space for doubt that spirituality is beneficial in such situations. The discussion and short conclusions sections add nothing new to that tenor.

Science is closely related to hypotheses - also in qualitative research, where the goal often lies in generating hypotheses. However, this seems not to be the case in the present research, which is implicitely oriented towards the hypothesis that spirituality is a powerful coping resource (note that this hypothesis is never explicitely mentioned). Given the implicit hypothesis, the analysis is entirely confirmatory. The authors report nothing that would be able to challenge the hypothesis, nor something that could specify it.

In the present form, I would not qualify the manuscript as scientific paper. However, I see starting points that might help improve it in that direction.

  1. Please provide the complete questions and/or guidelines for the interviews.
  2. Describe closely how you created your codes. Were there only codes related to spirituality? If so, this would be a severe shortcoming.
  3. Discuss surrounding conditions of the results. Could it be that your participants were somehow selected based on spirituality (e.g. by the snowball system)?
  4. Provide some quantitative information, which seems possible with your coding scheme. Many readers may like to know what proportion of all statements were related to spirituality.
  5. Discuss what your analyses add to the current state of research.

Some further issues:

  • The English needs some improvement. I've marked some flaws in the manuscript.
  • The life situation of the families is qualified as crisis. I don't think this notion is adequate, because a crisis is a short-term condition. I would agree that getting a child with DS may be a critical event - but the mean age of your children is three years. Weren't some parents accustomed to the situation? 
  • Table 1 contains no standard deviations

Reviewer 2 Report

This qualitative study is based on in-depth interviews collected among Brazilian parents having a child with Down syndrome. The authors were interested in the use of spirituality as a coping mechanism that can help parents to embrace their life with a disabled child, but also the nurses, who work with the families. The results show that spirituality was important to the parents, who were thus able to understand and accept their situation with a disabled child.

As the authors have made a secondary analysis of an existing qualitative data, they have focused on the aspects of spirituality in the interviews. It would help the readers if the readers would write down their research questions. The authors cannot expect from the readers that they know what is the "Model of Resilience, Stress, Adjustment and Family Adaptation". The research questions should be included in the introduction.

It is also important to tell the readers what are the main religions in Brazil. The interviewees refer to God: are they Christians/Catholics/Protestants/or do they belong to another church? Religion and faith seem to be important to the interviewees: what about other forms of spirituality?  

These research questions would help to understand how the authors discovered their initial codes. The authors explain well their method, but codes cannot be found "just like that" even in an inductive manner. Which codes were discarded, and were the interviews equally informative?

The authors use illustrative codes in Table 1. This is fine, but why use the same excerpts in the results? The results should be related to previous studies and to some kind of theoretical framework. Now there too many excerpts from interviews and too little theory. The result section must be re-written. 

In the discussion, the authors should not take spirituality for granted. What if parents believe in prayers and in spirits, but do not listen to health personnel? Is their spirituality helping the children or is it a coping mechanism for the parents? How are disabled children welcomed in the Brazilian society in general? How about the attitudes towards disabled children in Brazilian health care? Do parents believe in health care, medication and science?

Reviewer 3 Report

QSA refers to using existing collected data to do different research. This is very interesting because it allows us to explore new dimensions not initially worked on. Good work.

I do, however, have some suggestions that I believe may increase the quality of this article:

  1. You should correct the spelling mistake on page 1 (penultimate line): you wrote maor but I believe it will be
  2. The authors report using The Consolidated Criteria for Reporting Qualitative Research (COREQ) but this is not entirely clear throughout the article. This should be clarified.
  3. You should improve the graphic quality of Figure 1.
  4. You should correct the spelling mistake on page 5 (second illustrative quote): you wrote …conditions na adult… but I believe it will be …conditions an adult…
  5. In Table 1 some themes have two illustrative quotes, others three, and others more. Why? Since all are equally relevant, why not use the same criteria (number).
  6. Also in table one, why didn't you define " Meeting the needs of love”? I think it is important to define what this means in the context of this study.
  7. In the last theme and definition, you mention Contribution to the transformation in life.all changes in parents’ life since the child was born you should replace the dot with ifen, as used in the previous topics.
  8. Still on this topic, you should correct the spelling mistake: you wrote …Ihad… but I believe it will be …I had…
  9. The numbering of the tables is repeated. There are two tables numbered Table 1. It should be corrected.
  10. Since this is a secondary analysis of data from a previous study, what does the Characterization of parents and children with DS bring to this study? (p. 9 and 10), where was it used? is it necessary and appropriate in this context? I leave you with these thoughts.
  11. On pages 11 and 16 you do not use numerical referencing [3] for Narayanasamy. I suggest a correction.

Round 2

Reviewer 1 Report

In their reply to my comments, the authors focused on the fact that the present report is based on a secondary analysis of existing data. That made me aware that the original study seems not to have been published as yet. The authors do not refer to pertinent publications at the places in the text where they report that fact. So the reader has almost no information about the original study. In my eyes this is a serious weakness that makes me skeptical. 

The authors have put more effort in responding to my comments in their letter than in improving the manuscript. For example, they list their interview questions in the response letter, but only a small selection of the questions in the text. It is not helpful to explain the reviewer that reporting a standard deviation is not usual in qualitative research while still having "standard deviation" as a heading in Table 2.  

This leads me to the observation that the manuscript is still in a scetchy state: I could not find an abstract; Figure 1 appears twice; lnaguage issues remaining

The categories (codes, themes) are still not defined sufficiently. Finally, the manuscript is still written in a manner that shows no distance to the conviction that spirituality is a powerful coping resource. There is no real discussion (see "challenging or specifying the hypothesis").
